# Consecutive prediction of adverse maternal outcomes of preeclampsia, using the PIERS-ML and fullPIERS models: A multicountry prospective observational study

**Guiyou Yang**[1‡], **Tünde Montgomery-Csobán**[2‡], **Wessel Ganzevoort**[3,4], **Sanne J. Gordijn**[5], **Kimberley Kavanagh**[2], **Paul Murray**[6], **Laura A. Magee**[7], **Henk Groen**[1‡*], **Peter von Dadelszen**[7‡]

1 Department of Epidemiology, University Medical Center Groningen, University of Groningen, Groningen, the Netherlands, 2 Department of Mathematics and Statistics, University of Strathclyde, Glasgow, United Kingdom, 3 Department of Obstetrics and Gynecology, Amsterdam University Medical Centers, location AMC, Amsterdam, the Netherlands, 4 Amsterdam Reproduction and Development Research Institute, Amsterdam, the Netherlands, 5 Department of Obstetrics and Gynecology, University Medical Center Groningen, University of Groningen, Groningen, the Netherlands, 6 Department of Electronic and Electrical Engineering, University of Strathclyde, Glasgow, United Kingdom, 7 Institute of Women and Children's Health, School of Life Course and Population Sciences, King's College London, London, United Kingdom

‡ GY and TM-C share first authorship on this work. HG and PD share last authorship on this work.
* h.groen01@umcg.nl

## Abstract

### Background

Preeclampsia is a potentially life-threatening pregnancy complication. Among women whose pregnancies are complicated by preeclampsia, the Preeclampsia Integrated Estimate of RiSk (PIERS) models (i.e., the PIERS Machine Learning [PIERS-ML] model, and the logistic regression-based fullPIERS model) accurately identify individuals at greatest or least risk of adverse maternal outcomes within 48 h following admission. Both models were developed and validated to be used as part of initial assessment. In the United Kingdom, the National Institute for Health and Care Excellence (NICE) recommends repeated use of such static models for ongoing assessment beyond the first 48 h. This study evaluated the models' performance during such consecutive prediction.

### Methods and findings

This multicountry prospective study used data of 8,843 women (32% white, 30% black, and 26% Asian) with a median age of 31 years. These women, admitted to maternity units in the Americas, sub-Saharan Africa, South Asia, Europe, and Oceania, were diagnosed with preeclampsia at a median gestational age of 35.79 weeks between year 2003 and 2016. The risk differentiation performance of the PIERS-ML and fullPIERS models were assessed for each day within a 2-week post-admission window. The PIERS adverse maternal outcome includes one or more of: death, end-organ complication (cardiorespiratory, renal, hepatic, etc.), or uteroplacental dysfunction (e.g., placental abruption). The main outcome measures

**Data Availability Statement:** We are not the primary data stewards for these datasets that have been shared exclusively with our team. Our policy follows a managed open-access approach through academic collaboration. Therefore, any data sharing requires formal agreements between the original sites and data requestors. The contact email address for the datasets used in this study is: ma-piers@strath.ac.uk.

**Funding:** TM-C is funded by the University of Strathclyde, through the STRADDLE (University of Strathclyde Diversity in Data Linkage) Centre for Doctoral Training. The PIERS datasets were primarily funded by operating grants from the Canadian Institutes of Health Research (Human development, Child and Youth Health) and the Bill & Melinda Gates Institute for Population and Reproductive Health. Funders had no role in the design, analyses, interpretation, or manuscript preparation for this study. The intellectual property claim regarding the PIERS-AI machine learning-based suite of models (see competing interests) does not pose a financial interest for this study.

**Competing interests:** PvD, KK, LAM, PM, and TM-C have an intellectual property claim regarding the PIERS-AI machine learning-based suite of models (PE962099GB). No other conflicts are declared.

**Abbreviations:** AUC-PRC, area under the precision-recall curve; AUC-ROC, area under receiver operating characteristic curve; CI, confidence interval; GDP, gross domestic product; ISSHP, International Society for the Study of Hypertension in Pregnancy; LR, likelihood ratio; NICE, National Institute for Health and Care Excellence; PIERS, Preeclampsia Integrated Estimate of RiSk; PIERS-ML, PIERS Machine Learning; ROC, receiver operating characteristic; SpO2, oxygen saturation.

were: trajectories of mean risk of each of the uncomplicated course and adverse outcome groups; daily area under the precision-recall curve (AUC-PRC); potential clinical impact (i.e., net benefit in decision curve analysis); dynamic shifts of multiple risk groups; and daily likelihood ratios. In the 2 weeks window, the number of daily outcome events decreased from over 200 to around 10. For both PIERS-ML and fullPIERS models, we observed consistently higher mean risk in the adverse outcome (versus uncomplicated course) group. The AUC-PRC values (0.2–0.4) of the fullPIERS model remained low (i.e., close to the daily fraction of adverse outcomes, indicating low discriminative capacity). The PIERS-ML model's AUC-PRC peaked on day 0 (0.65), and notably decreased thereafter. When categorizing women into multiple risk groups, the PIERS-ML model generally showed good rule-in capacity for the "very high" risk group, with positive likelihood ratio values ranging from 70.99 to infinity, and good rule-out capacity for the "very low" risk group where most negative likelihood ratio values were 0. However, performance declined notably for other risk groups beyond 48 h. Decision curve analysis revealed a diminishing advantage for treatment guided by both models over time. The main limitation of this study is that the baseline performance of the PIERS-ML model was assessed on its development data; however, its baseline performance has also undergone external evaluation.

## Conclusions

In this study, we have evaluated the performance of the fullPIERS and PIERS-ML models for consecutive prediction. We observed deteriorating performance of both models over time. We recommend using the models for consecutive prediction with greater caution and interpreting predictions with increasing uncertainty as the pregnancy progresses. For clinical practice, models should be adapted to retain accuracy when deployed serially. The performance of future models can be compared with the results of this study to quantify their added value.

---

## Author summary

### Why was this study done?

- The fullPIERS model was considered to be the best model for predicting the adverse outcomes of preeclampsia within 2 days following first admission, until recently when the PIERS-ML model was introduced.

- The National Institute of Health and Care Excellence guideline in the United Kingdom recommends using the fullPIERS model to make serial predictions; however, to the best of our knowledge, this model has never been verified for this use.

### What did the researchers do and find?

- Using the largest available data set in this field (to the best of our knowledge), we quantified the performance of both the fullPIERS and PIERS-ML models (as PIERS-ML has even higher predictive accuracy) when they are used for serial predictions.

- We found that the performance of both models deteriorated considerably over time in the first 2 weeks following admission to hospital.

**What do these findings mean?**

- In the absence of models that have been validated for serial predictions of adverse maternal outcomes of preeclampsia, clinicians may still use fullPIERS and PIERS-ML models for ongoing assessments after the first admission with preeclampsia, but the predictions should be treated with increasing caution as the pregnancy progresses.

- More research is needed to develop models that perform well over time when used for repeated predictions.

- The main limitation of this study is that we assessed the initial performance of the PIERS-ML model on the same data used to create it. However, the model's initial performance has also been tested on data from outside sources.

## Introduction

Of women whose pregnancies are complicated by preeclampsia, 5% to 20% will develop severe complications, particularly if the syndrome is of early onset (e.g., <34 weeks gestation) [1,2]. Both over- and under-treatment—iatrogenic harm from prematurity and increased healthcare costs—are potential consequences of inaccurate prediction of severe complications.

The existing prediction models for severe complications of preeclampsia were developed primarily utilizing baseline information [3–5]. Notably, the fullPIERS (Preeclampsia Integrated Estimate of RiSk) model was developed from data from well-resourced settings to predict the risk of the adverse maternal outcomes of both early- and late-onset preeclampsia, within 48 h following hospital admission [5]. This interval is clinically useful, as it reflects the opportunity to arrange in utero transfer, induce labor, and achieve the full benefit of antenatal corticosteroids for fetal lung maturation, as relevant [6]. The recent PIERS-ML (machine learning) model shares the fullPIERS model's objectives, but distinguishes itself by being trained on data from globally diverse settings [7].

To be useful during ongoing clinical care, particularly during expectant management of preterm preeclampsia [1], risk prediction needs to be updated regularly, to monitor disease risk progression as clinicians otherwise do informally. In the absence of a preeclampsia outcome prediction model that can accommodate repeated measurements to guide joint decision-making by women with preeclampsia and their maternity care providers, the National Institute of Health and Care Excellence guideline in the United Kingdom suggests that the fullPIERS model be used iteratively for consecutive prediction for the same woman [8]. As an updated version of the fullPIERS model, PIERS-ML may well be employed for consecutive prediction in the same way. Regardless, neither PIERS-ML nor fullPIERS has been validated for such application.

This study aimed to evaluate whether or not the performance of the PIERS-ML and fullPIERS models justifies their application for consecutive prediction of adverse maternal outcome in preeclampsia.

## Methods

### Study design and data source

In this prospective observational study, we utilized the PIERS-ML and fullPIERS models [5,7], and a pooled database of 8,843 women diagnosed with preeclampsia at a median gestational age of 35.79 (interquartile range: 31.79 to 38.23) weeks. Data were collected between year 2003 and 2016 from maternity units in the Americas, sub-Saharan Africa, South Asia, Europe, and Oceania. We chose this pooled database for its global scope and large sample size, which enhances the generalizability of results. The baseline data of this database was used for development of the PIERS-ML model [7]. This database includes both the development and external validation cohorts for the miniPIERS (for low- and middle-income countries) and fullPIERS models (for high-income countries) [5,9–12]. We assessed the performance of both the PIERS-ML and fullPIERS models for consecutive prediction of adverse maternal outcome using the baseline data and follow-up measurements of this pooled database. This study was approved by the NHS Research Ethics Committee (REC reference: 02-03-033 on 11 March 2003). Written informed consent was obtained from all participants apart from those recruited from site Canada where the data were acquired through an audit. This study was conducted according to the guidelines of the Declaration of Helsinki and reported as per the Strengthening the Reporting of Observational Studies in Epidemiology (STROBE) guideline (S1 Checklist).

### Inclusion and exclusion criteria

We used prospectively collected data from women with preeclampsia, broadly defined according to the 2021 International Society for the Study of Hypertension in Pregnancy (ISSHP) criteria [13]. Follow-up was performed via the routine prenatal and postnatal clinical visits of women. The follow-up measurements were not used once the patient developed any component of the combined adverse maternal outcome.

Sites that contributed data to this combined cohort had a general policy of expectant management for women with preterm preeclampsia and, post-HYPITAT study [14], a general policy of induction at term.

### The PIERS combined adverse maternal outcome

The primary study outcome was a composite developed by Delphi consensus [15], and defined as one or more of the following within a 2-day rolling window: (i) maternal mortality; (ii) severe maternal morbidity affecting the central nervous, cardiovascular, respiratory, renal, hepatic, or hematologic systems; or (iii) other serious complications, such as placental abruption (listed in S1 Table). This combined outcome is similar to (but not completely consistent with) the more recent Delphi-derived iHOPE core maternal outcome set for women with preeclampsia [16]. Women who did not develop the composite adverse maternal outcome were defined as having had an "uncomplicated course."

### Prediction models

The predictors in the PIERS-ML and fullPIERS models [5,7], and the distribution of events for each component of the composite outcome within the pooled database, are presented in S2 Table. The measurements of the predictors and assessment of the outcome components were conducted according to the clinical guideline at each site. Quantitative variables, such as platelet count and gestational age on admission, were retained in their original form in both models instead of being categorized.

## Statistical analyses

The predicted probability of the PIERS-ML model may be used to dichotomize women into high- or low-risk groups, or to stratify them into multiple data-driven risk groups based on likelihood ratios (LRs) [7]. Thus, we explored the performance of the model when used for consecutive prediction for each of these risk stratification scenarios.

The date of first assessment with preeclampsia was set as day 0 in this study. Using the latest measurements available each day, we re-calculated the predicted probability of developing an adverse maternal outcome within 48 h based on the PIERS-ML model. This enabled us to evaluate a 2-week trajectory (i.e., days 0 to 13) of predicted probabilities. A duration of 2 weeks was chosen because over 98% of the adverse maternal outcomes in preeclampsia occurred over this timeframe, and it was considered a sufficient time window to evaluate model performance.

We plotted the trajectory of the mean predicted probabilities and the number of patients with measurements for both the "uncomplicated course" and adverse outcome groups, respectively. This step was conducted to show the change in the mean predicted probabilities in each group, as well as the difference between the groups.

## Use of the PIERS-ML model with patients dichotomized into high versus low-risk

The difference in mean predicted probabilities between the "uncomplicated course" and adverse outcome groups does not reliably indicate the discriminative capacity of the model. Typically, to assess this aspect, a receiver operating characteristic curve (ROC curve) is employed. However, due to the relatively low number of adverse outcomes, the predominance of negative cases can lead to a relatively high area under the ROC curve (AUC-ROC) despite poor sensitivity (S1 Fig). To counteract this, we measured performance using the area under the precision recall curve (AUC-PRC) value for each day (day 0 to day 13) to show how the model's discriminative ability changed over time [7]. The AUC-PRC value, ranging from 0 to 1, is a measure of model effectiveness based on the balance between positive predictive value (the probability that a patient flagged as high risk by the tool actually has an adverse outcome, termed as "precision" in this measure) and sensitivity (the probability that an individual with an adverse outcome is flagged as high risk, termed as "recall"). While AUC-ROC values have a fixed baseline value (0.5) for comparison, AUC-PRC values are compared to the fraction of positives as a baseline, which is the number of cases with adverse outcomes divided by the total participants [17]. A higher AUC-PRC value indicates superior model performance in accurately identifying positive cases while minimizing false positives. In the case of data sets with a low incidence of adverse outcomes, a low value of AUC-PRC can still indicate good model performance, unlike AUC-ROC, if the value is higher than the fraction of positives.

## Use of the PIERS-ML model with patients stratified into multiple risk groups

Based on the magnitude of positive likelihood ratios (used for the high-risk groups; indicating how much more likely a true-positive individual is flagged as high risk compared to a true-negative individual) and negative likelihood ratios (used for the low-risk groups; indicating how much more likely a true-positive individual is flagged as low risk compared to a true-negative individual) of the predicted risks, the PIERS-ML model categorizes women into 5 risk groups: very low risk (negative LR [-LR] <0.1), low risk (-LR 0.1–0.19), moderate risk (LR 0.2–5.0), high risk (positive LR [+LR] 5.1–10), and very high risk (+LR >10) with the following cut-off

values of the predicted risks: <0.6%, 0.6% to 3.1%, >3.1% to <18.8%, 18.8% to 45.6%, and >45.6% [7]. To evaluate the change in the potential clinical impact of the model over an extended period, a series of decision curve analyses (on day 0, day 4, day 8, and day 13) was developed. Net benefit was the measure of the potential clinical impact, as recommended in the TRIPOD (Transparent Reporting of a multivariable prediction model for Individual Prognosis or Diagnosis) guideline [18], and threshold probability (cut-off value) is assumed to be informative of how a clinician or patient weighs the relative harm of a false positive versus a false negative [19]. A decision curve plots net benefit on the y-axis against threshold probability on the x-axis, and by default, compares the model under study to "treat all" and "treat none" policies [19–21]. In the context of this study, the "treat all" policy refers to immediate delivery for all patients, whereas "treat none" means expectant management for all patients. The x-axis was limited to the range of 0% to 60%, as this range was considered sufficient to cover the clinically plausible threshold probabilities at which patients or clinicians would opt for intervention in this study, and the threshold probabilities for risk stratification by the PIERS-ML model (i.e., 0.6%, 3.1%, 18.8%, and 45.6%) were marked on the x-axis.

A Sankey diagram was utilized to provide an overview of the dynamic shifts in risk groups and their respective contributions to adverse outcomes or an uncomplicated course when the model was used for consecutive prediction. The width of "paths" in a Sankey diagram is proportional to the number of participants [22].

Lastly, we quantified the daily +LR by applying predefined thresholds for both high-risk and very high-risk groups, to illustrate how the model's "rule-in" ability evolves over time. Similarly, we computed the daily -LR using thresholds for the low risk and very low-risk groups to demonstrate how the model's "rule-out" capacity changes over time.

### Performance of the fullPIERS model for consecutive prediction

The fullPIERS model, first published in 2011, was developed to dichotomize patients into "high-risk" and "not high-risk" but in the setting of developed countries [5]. We evaluated its performance deployed for consecutive prediction by quantifying the change of AUC-PRC (AUC-ROC values shown on S4 Fig for comparison), plotting the trajectory of mean predicted risks and decision curves in this study. The results can be found in the Supporting information (S2–S5 Figs).

The percentage of missing values was around 18%. Multiple imputation (20 times) method was used to address data missing at random or completely at random, while the last observation carried forward method was employed for data missing not at random, such as platelet counts, where clinicians may skip measurements if biomarkers are assumed unchanged. We had verbal agreement on the analysis plan in February 2022, and added calculation of AUC-PRC later based on the reviewer comments on the PIERS-ML model manuscript [7]. Analyses were performed using R Statistical Software (version 4.0.5, R Foundation for Statistical Computing, Vienna, Austria), R Studio (version 1.4.1106), and the following packages: tidyverse (version 2.0.0), MICE (version 3.16.0), easyalluvial (version 0.3.1), zoo (version 1.8–11), ggsankey (version 1.0), magrittr (version 2.0.3), epiR (version 2.0.63), pROC (version 1.18.0) [23], and dcurves (0.4.0).

## Results

### Baseline characteristics of women and missingness of variables

As shown in Table 1, data of 8,843 women with preeclampsia across various continents were analyzed in this study, 32% of them being white, 30% black, and 26% Asian. Over one third (38.1%) of the participants were recruited from Canada. The median (interquartile range) of

**Table 1. Characteristics of women at admission.**

| Variables at admission | Baseline on day of admission (*N* = 8,843; No. of adverse outcome: 1,083) | Number (%) of women with at least 1 missing value |
|---|---|---|
| **Time from admission to final discharge (days)** | 5 [3–11] | - |
| *Country* | - | 0 (0%) |
| Australia | 245 (2.77%) | - |
| Brazil | 188 (2.13%) | - |
| Canada | 3,369 (38.1%) | - |
| Fiji | 136 (1.54%) | - |
| Finland | 119 (1.35%) | - |
| New Zealand | 340 (3.85%) | - |
| North America | 1,244 (14.07%) | - |
| Pakistan | 1,083 (12.25%) | - |
| South Africa | 346 (3.91%) | - |
| *Health system* | | |
| GDP per capita (US$) | 43,585.51 [26,869.67–50,114.18] | 0 (0%) |
| National m**aternal mortality ratio (per 100,000 live births)** | 11 [11.00–15.00] | 0 (0%) |
| *Demographics* | | |
| Ethnicity | - | 1,025 (11.59%) |
| White | 2,832 (32.02%) | - |
| Black | 2,684 (30.36%) | - |
| Asian | 2,304 (26.06%) | - |
| Others | 1,023 (11.57%) | - |
| **Age of patient at estimated** due date (years) | 31 [26.00–36.00] | 21 (0.24%) |
| Height on admission (cm) | 162.5 [157.48–167.27] | 884 (10.00%) |
| **Gestational age on eligibility (weeks)** | 35.79 [31.79–38.23] | 17 (0.19%) |
| *Symptoms or lab parameters* | | |
| **Symptom of chest pain or** dyspnea | 186 (2.10%) | 3,347 (37.85%) |
| **Highest systolic blood pressure (mmHg)** | 151.78 [140–161.35] | 1,323 (14.96%) |
| **Highest diastolic blood pressure (mmHg)** | 96 [90–100.6] | 1,324 (14.97%) |
| SpO2 (%) | 97 [96.85–97] | 987 (11.16%) |
| Haematocrit (%) | 0.36 [0.34–0.38] | 4,111 (46.49%) |
| **Total leucocyte count (× 109 per L)** | 10.66 [9.32–12] | 3,189 (36.07%) |
| **Platelet count (× 109 per L)** | 210 [173–244] | 1,820 (20.58%) |
| Mean platelet volume (fL) | 9.99 [9.09–11.1] | 5,135 (58.08%) |
| Serum creatinine (μmol/L) | 60.15 [52.21–70] | 2,481 (28.06%) |
| Uric acid (mmol/L) | 342 [297.4–389.99] | 3,147 (35.59%) |
| Aspartate transaminase (U/L) | 30 [23.00–43.05] | 3,629 (41.04%) |
| Alanine transaminase (U/L) | 24 [15.00–37.20] | 2,552 (28.86%) |
| Serum albumin (g/L) | 28.8 [22.55–32.00] | 4,823 (54.55%) |

Data are *n* (%) or median [Interquartile range]. Ethnicity has been defined in S1 Table.

GDP: gross domestic product; SpO2: Oxygen saturation.

the follow up time was 5 (3 to 11) days. In total, 1,083 developed the defined adverse outcome event. The median age of the women at estimated due date was 31 (interquartile range: 26 to 36) years. The median gestational age at admission was 35.79 (interquartile range: 31.79 to 38.23) weeks. Around 10% of the women had missing value of height, and 20% to 58% of the women had at least 1 missing value of each laboratory test during the 2-week window.

### Trajectory of mean predicted probabilities and number of patients with measurements

A consistent decline was seen in the number of patients with measurements from day 0 to day 13 following admission (Bars, Fig 1). The PIERS-ML-derived mean predicted probability of adverse outcomes in the next 48 h within the adverse outcome group peaked on day 0, steadily decreased until day 3, and then fluctuated between 0.1 and 0.2 (pink line, Fig 1); the number of women in this group was quite small (around 300) from day 3 and kept decreasing (pink bars, Fig 1). Conversely, the mean predicted probability of adverse outcomes in the uncomplicated-

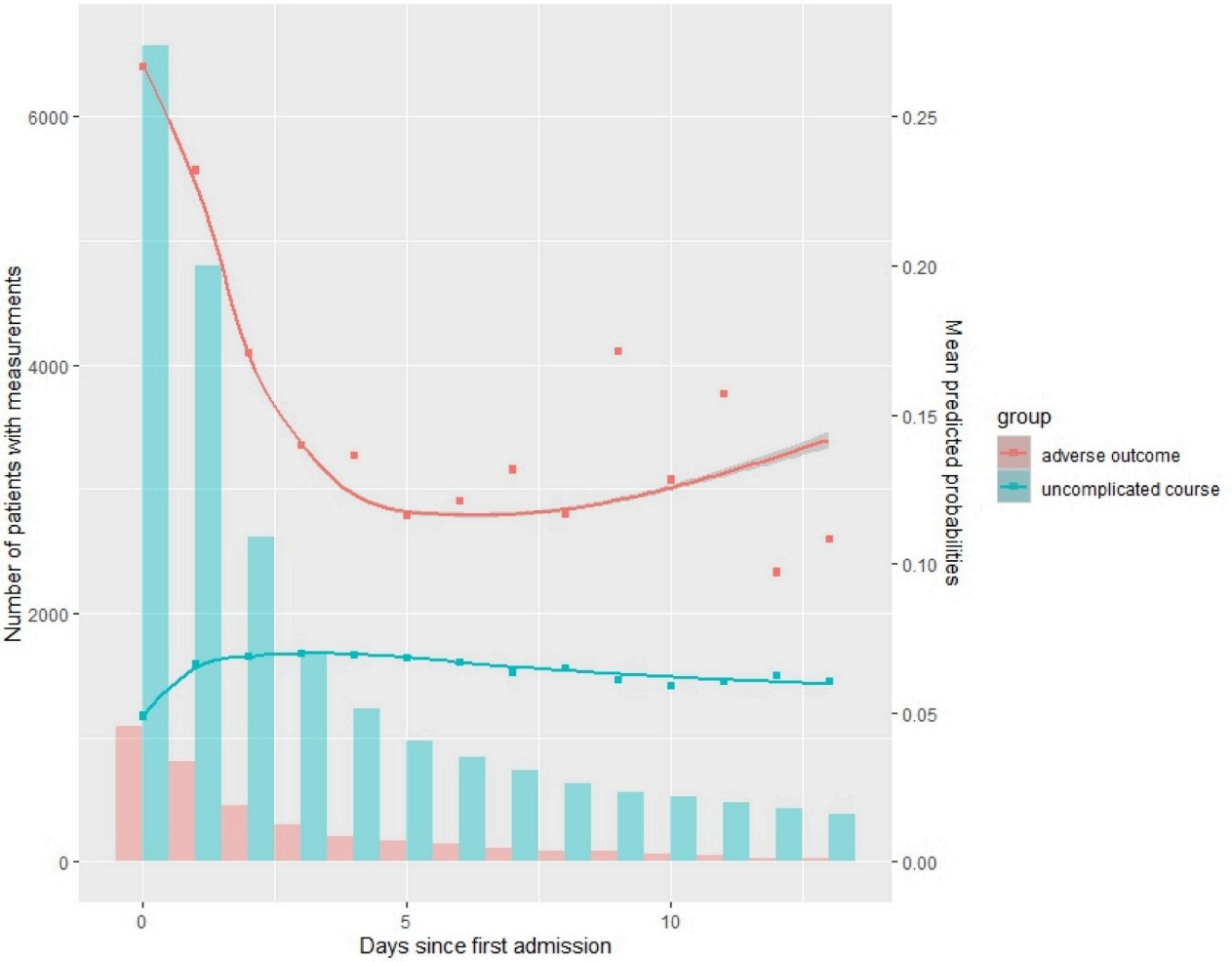

**Fig 1.** Mean predicted probabilities of complications in the next 48 h (lines) and number of patients with measurements (bars) in uncomplicated course (blue) and adverse outcome group (pink) per day since admission using the PIERS-ML model. PIERS-ML, PIERS Machine Learning.

course group was at its lowest on day 0, experienced an upward trend until day 2, and subsequently remained stable (blue line, Fig 1). The adverse outcome group consistently exhibited a higher mean predicted probability than the uncomplicated course group. The mean of the probabilities predicted by the fullPIERS model (S2 Fig) exhibited similar patterns of consistently higher values in the adverse outcome group with fluctuation over time, while the uncomplicated course group had lower and more stable values.

## Area under the precision-recall curve per day

The PIERS-ML AUC-PRC values on day 0 and day 1 were 0.65 and 0.52, respectively, but thereafter, all AUC-PRC values were within 0.1 to 0.5, and the lower boundary of the 95% confidence interval (95% CI) of the AUC-PRC value overlapped with the fraction of adverse outcomes on day 12 and day 13 (Fig 2, black line). The 95% CI became wider from day 0 to day 13 as the number of daily outcome events decreased from over 200 to around 10 (Fig 2, bars). Although the AUC-PRC values stayed mostly above the corresponding values of the fraction of adverse outcomes (Fig 2, pink line) within 2 days, the gap between the 2 significantly shortened over time. Conversely, for the fullPIERS model (S3 Fig), the AUC-PRC values fell between 0.2 and 0.4 on most days (black line) without a clear pattern of deterioration, but shared a similar pattern with the PIERS-ML model regarding the progressively widening 95% CIs as the count of outcomes fell (bars), and relationship with the fraction of adverse outcomes within 2 days (S3 Fig pink line).

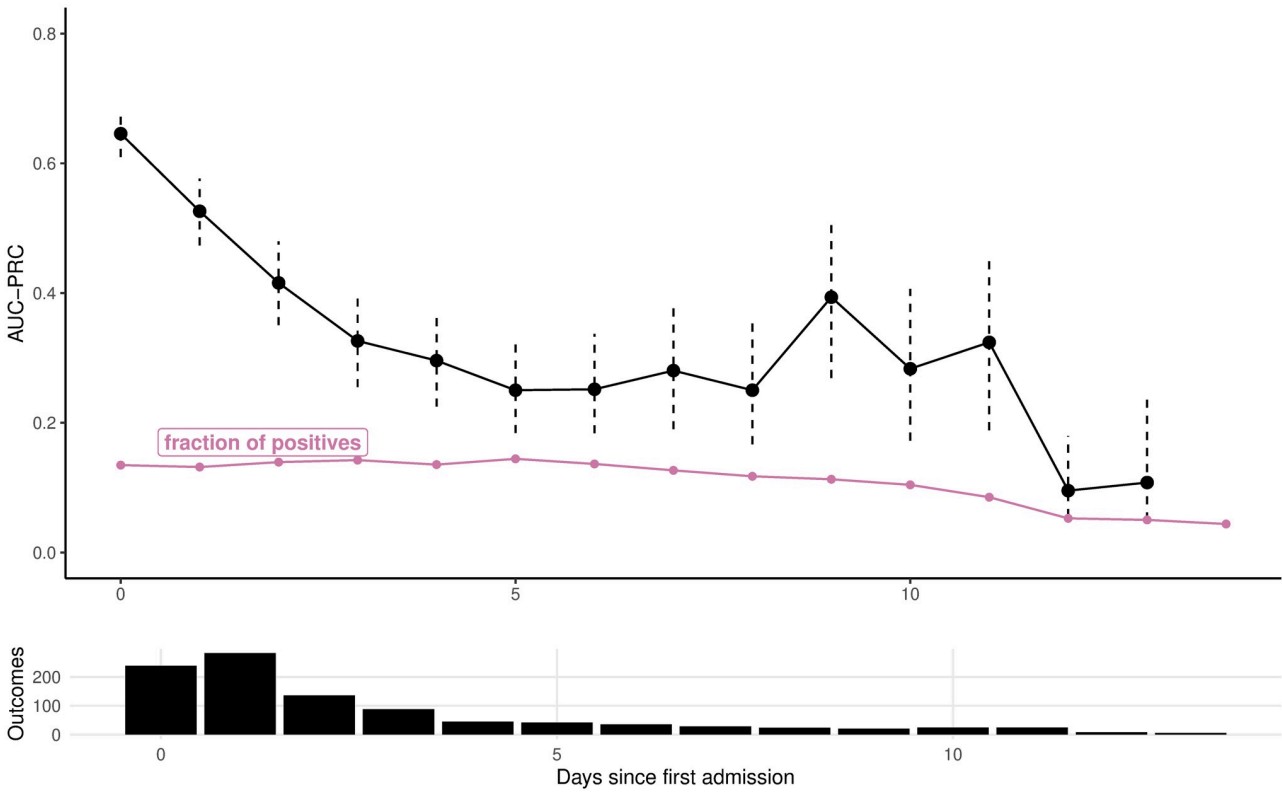

**Fig 2. Daily number of adverse outcomes (bars) and area under the precision-recall curve (AUC-PRC) per day of the PIERS-ML model.** Dashed vertical lines indicate 95% CIs. Fraction of adverse outcomes within 2 days is calculated as the number of women that would experience adverse outcomes within 2 days divided by the total number of remaining women on a certain day. AUC-PRC, area under the precision-recall curve; CI, confidence interval; PIERS-ML, PIERS Machine Learning.

S1 Text provides more information for interpreting AUC-PRC values.

## Decision curve analyses

Fig 3 presents the results of PIERS-ML-based decision curve analysis on days 0 (Fig 3a), 4 (Fig 3b), 8 (Fig 3c), and 13 (Fig 3d). The dashed vertical lines represent 4 threshold probabilities (i.e., 0.6%, 3.1%, 18.8%, and 45.6%) from the PIERS-ML model [7] that stratify patients into very low risk (<0.6%), low-risk (0.6% to 3.1%), moderate risk (>3.1% to <18.8%), high-risk (18.8% to 45.6%), and very high-risk (>45.6%) groups. The dashed orange line represents a

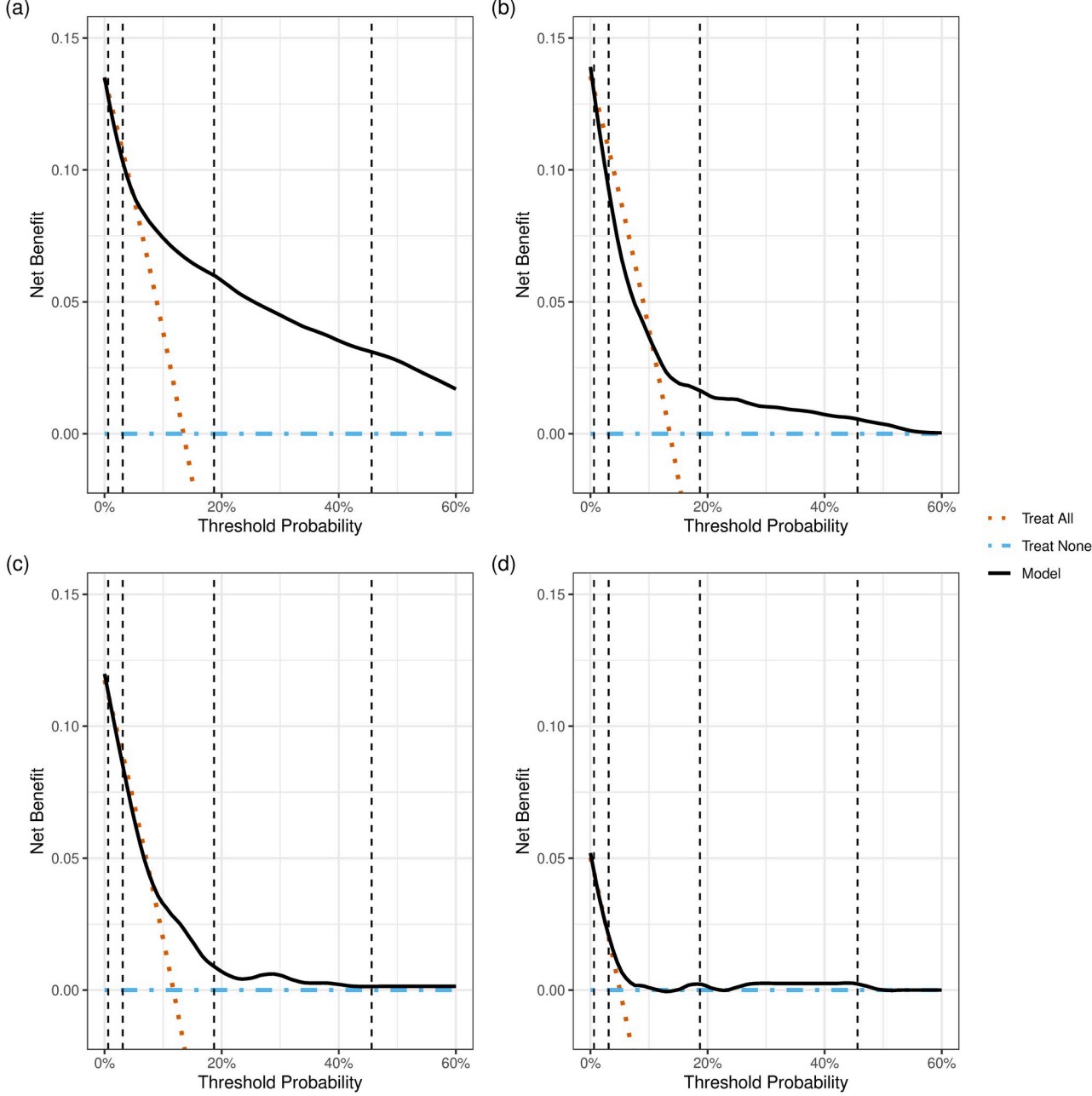

**Fig 3. Decision curve analysis of PIERS-ML model on day 0 (a), day 4 (b), day 8 (c), and day 13 (d).** PIERS-ML, PIERS Machine Learning.

"treat all" approach, the dashed blue line a "treat none" approach, and the solid black line the PIERS-ML model performance. Net benefit of PIERS-ML is reflected by the solid black line being above the dashed lines for the other approaches.

Dashed vertical lines (from left to right) represent 4 threshold probabilities, i.e., 0.6%, 3.1%, 18.8%, and 45.6%, respectively, that stratify patients into very low-risk, low-risk, moderate risk, high-risk, and very high-risk groups. Net benefit indicates clinical usefulness of a model. At a chosen threshold probability, a higher net benefit indicates a better model. Here, we show the serial comparison between using the PIERS-ML model and scenarios without a prediction model (either "treat all" or "treat none").

The benefit of the PIERS-ML model was greatest for those flagged as high or very high risk (Fig 3). On each day, when the threshold probability of receiving treatment was 0.6%, treatment guided by the model was predicted to be as good as treating all patients. When the threshold probability was 3.1%, the net benefit of treatment guided by the model was generally predicted to be very close to that of treating all patients. However, this benefit was predicted to be inferior to the approach of treating all patients on day 4 (Fig 3b). When the threshold probability was 18.8% or 45.6%, treatment guided by the model was predicted to result in higher net benefit (especially on day 0: Fig 3a and day 4: Fig 3b), but relative advantage decreased over time.

Treatment guided by the fullPIERS model (S5 Fig) was predicted to have a small advantage over treating all patients on day 0 (S5a Fig); however, due to rapid deterioration of the net benefit, there was almost no difference between treatment guided by the fullPIERS model and the "treat all" policy beyond day of admission, on days 4 (S5b Fig), 8 (S5c Fig), or 13 (S5d Fig).

S1 Text provides more explanation to interpreting the results of decision curve analysis.

## Dynamic shifts of risk groups under consecutive prediction

The Sankey diagram (Fig 4) provides a descriptive overview of the dynamic shifts of the 5 risk groups when the PIERS-ML model was used repeatedly over days. It reveals that patients of moderate risk (in yellow) constituted the largest daily proportion. In general, the top 3 risk groups were the primary contributors to adverse events, while in the low risk and very low-risk groups, adverse events were rare.

Of the 7,600 patients, 327 (4.3%) were predicted to be at very high risk of an adverse outcome at least once in the 2 weeks following first assessment; 316/327 (96.6%) experienced their adverse outcome within 2 days of first being classified as very high risk. Only 11/327 (3.4%) had an uncomplicated course, with 10 having been delivered on the day that they were first being predicted to be at very high risk.

A total of 860 (11.3%) patients were predicted to be at high risk of an adverse outcome at least once, and 426/860 patients (49.5%) experienced an adverse outcome, 39 of whom moved to the very high-risk group before the adverse outcome occurred. Most adverse outcomes (388/426, 91.1%) occurred within 2 days of classification into high risk for the first time, and only 8 patients (1.9%) experienced an adverse outcome after more than a week. Approximately half of patients (434/860, 50.5%) had an uncomplicated course, most delivering within 2 days of first being predicted to be at high risk (358/434, 82.5%); however, some (76/434, 17.5%) remained under expectant management with no complications for up to 2 weeks.

For patients in the moderate risk group at admission, nearly half delivered within 2 days following admission; after day one, most patients remained in the moderate risk group on the subsequent day, but on each day thereafter, there were similar proportions of patients switching to the high-risk or low-risk groups.

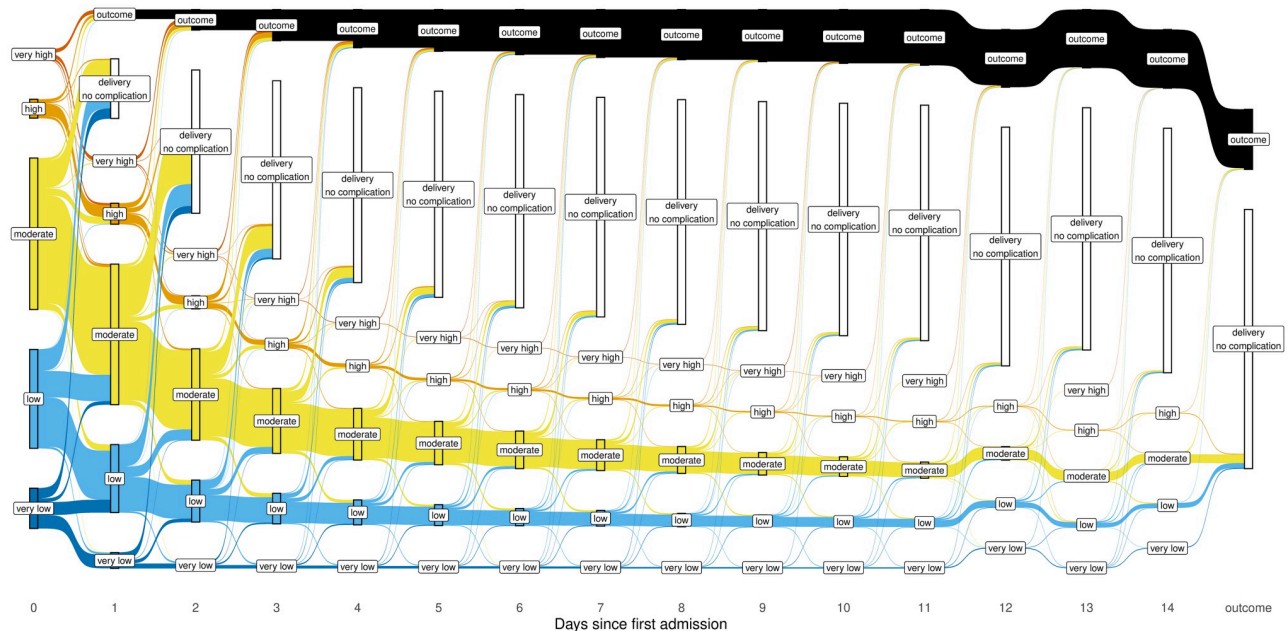

**Fig 4. Sankey diagram showing an overview of the dynamic shifts of the 5 risk groups from day 0 to day 14 after admission.** The flow (demonstrating the shifts among the risk groups) of each risk group is featured by a distinctive color (e.g., dark blue for the very low risk group). The white bars show the cumulative deliveries over time, while the black band represents the cumulative adverse outcome events.

A total of 3,147 patients (35.6%) were predicted to fall into the low-risk group at least once over the 2-week period. Only a small proportion (159/3,147, 5.1%) experienced an adverse outcome, but most (114/159, 71.7%) moved to moderate, high, and/or very high risk before occurrence of that outcome, and 2,988 patients (94.9%) had an uncomplicated course; most (2,636/2,988, 88.2%) delivered within a week of first being predicted to be at low risk; however, some remained under expectant management with no complications for up to 2 weeks.

Of the 1,125 patients classified as being at very low risk at least once, 23 (2.0%) suffered an adverse outcome, 21 of whom moved to a higher risk category before occurrence of that outcome, and 1,102 (98.0%) had an uncomplicated course, with 803/1,102 (72.9%) delivering within 2 days, and some remaining under expectant management with no complications for up to 2 weeks.

S1 Text provides more information for interpreting the Sankey diagram.

## Change of likelihood ratios

Table 2 shows the change of +LRs and -LRs across different risk groups from day 0 to day 13. In the very high-risk group, the +LR consistently exhibited high values, from 70.99 to infinity (except for day 12 when no women were identified as being at very high risk). For the high-risk group, the +LR exceeded 10 during the initial 48 h, and then fluctuated from 2.01 to 7.92 thereafter. The -LRs for the low-risk group were below 0.05 for the first 48 h, then fluctuated between 0.15 and 0.80 thereafter. For the very low-risk group, the -LRs were predominantly 0 throughout.

## Discussion

In this multicountry prospective observational cohort study of 8,843 pregnancies, we found that neither the PIERS-ML nor fullPIERS model maintained good performance employed for

**Table 2. Daily positive/negative likelihood ratios calculated with thresholds of different risk groups.**

| Days | Very high-risk[$] | High-risk[$] | Low-risk[*] | Very low-risk[*] |
|---|---|---|---|---|
| 0 | 70.99 (51.55, 97.77) | 16.44 (13.89, 19.46) | 0.04 (0.02, 0.08) | 0.00 |
| 1 | 173.99 (85.77, 352.95) | 11.73 (10.16, 13.53) | 0.03 (0.01, 0.07) | 0.00 |
| 2 | 196.70 (47.59, 813.01) | 7.92 (6.36, 9.88) | 0.19 (0.12, 0.31) | 0.00 |
| 3 | 174.23 (23.07, 1,315.71) | 6.68 (5.01, 8.91) | 0.29 (0.18, 0.48) | 0.54 (0.07, 3.98) |
| 4 | Inf | 5.99 (4.03, 8.90) | 0.45 (0.27, 0.75) | 0.00 |
| 5 | 96.18 (11.99, 771.80) | 3.62 (2.03, 6.48) | 0.54 (0.33, 0.87) | 0.00 |
| 6 | Inf | 4.14 (2.25, 7.63) | 0.49 (0.27, 0.87) | 0.00 |
| 7 | Inf | 4.56 (2.49, 8.37) | 0.30 (0.14, 0.64) | 1.56 (0.20, 12.12) |
| 8 | Inf | 3.92 (1.93, 7.99) | 0.19 (0.06, 0.56) | 0.00 |
| 9 | Inf | 6.00 (2.96, 12.17) | 0.15 (0.05, 0.46) | 0.00 |
| 10 | Inf | 3.32 (1.30, 8.45) | 0.21 (0.08, 0.54) | 0.00 |
| 11 | Inf | 5.42 (2.37, 12.38) | 0.24 (0.08, 0.71) | 0.00 |
| 12 | NA | 2.01 (0.29, 13.98) | 0.80 (0.35, 1.82) | 0.00 |
| 13 | Inf | 3.63 (0.86, 15.22) | 0.49 (0.20, 1.20) | 0.00 |

Numbers in "()": 95% confidence intervals.

[$]Positive likelihood ratio.

[*]Negative likelihood ratio.

Inf, infinite; NA, not applicable.

repeated risk stratification in women with preeclampsia. This was true over a number of analytical approaches: trajectories of mean predicted probabilities, area under the precision-recall curve per day, decision curve analyses, dynamic shifts of risk groups under consecutive prediction, and change of LRs. While the PIERS-ML model displayed good "rule-in" and "rule-out" capacity for the very high-risk and very low-risk groups over time, performance of the much larger high-risk and low-risk groups deteriorated substantially after 48 h following admission, when using the latest available measurements.

In our study, we have quantified the performance of the PIERS-ML and fullPIERS models for consecutive prediction. To the best of our knowledge, we could not identify any existing studies that assessed static prediction models employed for consecutive prediction, much less prediction models specifically in the context of preeclampsia. Clinical periodic assessments are increasingly integral to patient care, rendering consecutive prediction both more common and important.

Consecutive predictions using both the PIERS-ML and fullPIERS models revealed disparity in the trajectories of the mean predicted probabilities in the adverse outcome group and the uncomplicated course group, which can be attributable to various external factors. Most adverse events occurred soon after admission, supporting the early performance of the models. Subsequently, early interventions in the sickest patients, and less severe conditions in the remaining patients, may have resulted in relatively lower predicted probabilities. As more births occurred, the rapidly decreasing and small number of patients contributed to notable variation in the estimated mean predicted probability. Conversely, those women with uncomplicated courses may have presented initially with milder disease and, thus, lower predicted probabilities. However, their condition deteriorated subsequently, resulting in an increased mean predicted probability and eventual stabilization due to medical intervention. Furthermore, both models' inability to capture the trajectory of predictors over time could be a pivotal

factor in the relatively steady pattern of mean predicted probability for both groups after 48 h following admission.

Although the AUC-PRC value of the PIERS-ML model was mostly above the fraction of the adverse outcomes within 2 days, it generally decreased over time. This reduction, along with the narrowing gap between the AUC-PRC value and the fraction of adverse outcomes, indicates a declining discriminative capacity. This decline may be attributed to the model's reliance on baseline, rather than repeated, measurements [7]. Although advancing gestational age could signify different severity, given that a longer duration allows for greater progression towards a maternal high-risk scenario, gestational age was included as a predictor in the model. Alternatively, the lack of consecutive prediction could lie in the static nature of the PIERS-ML model, which essentially captures a "snapshot" of a patient's condition at first assessment. In contrast, clinical practice involves regular reassessment to monitor disease progression—individual trajectories of predictors.

It is noteworthy that some +LRs of the very high-risk group were infinite, which resulted from all women in that group experiencing an outcome. However, the patient numbers in these groups were quite small, and so may not reflect reliable discriminative capacity. In general, both the descriptive Sankey diagram and the daily LRs demonstrated that the very low-risk group and the very high-risk group maintained good "rule-out" and "rule-in" capacity over time, respectively, but the very high- and very low-risk groups accounted for only a small proportion of the patients. In contrast, the high- and low-risk groups had many more patients and the predictive accuracy decreased substantially after 48 h following admission, making the deterioration in the overall performance of the model over time clinically relevant.

The results of decision curve analysis, which incorporates how the harm of a false positive prediction weighs against that of a false negative prediction, aligned with the findings of other analyses in this study. Moreover, the time and effort needed to gather data for and implement a model are not considered in decision curve analysis. Thus, if predictors of the model take non-trivial efforts, the model would not be considered truly useful if it only brings a slight increase in net benefit [20]. This means the model under study might be not helpful when employed, e.g., on day 8 or day 13 considering its small advantage in terms of net benefit. The change of the net benefit of the model could be explained by similar reasons responsible for the trajectories of the mean risk and the change of AUC-PRC values described above.

The fullPIERS model underperformed in this study due to several factors. Firstly, the model's static nature proved unsuitable for consecutive predictions as discussed above. Secondly, it was tested on a more geographically diverse data set than the one used for training, which may have altered its accuracy and contributed to its inferiority compared to the PIERS-ML model. Additionally, AUC-PRC was not reported in the original study. Although the AUC-ROC value was previously reported, it is not directly comparable to AUC-PRC and is generally not an accurate measure of model performance when positive cases are rare [5]. Strengths of our study include the large sample size with diverse settings and comprehensive approach, examining PIERS models derived using machine-learning and multiple regression, and taking multiple analytical approaches. A limitation of this study is that the baseline performance (day 0) of the PIERS-ML model, as a benchmark for comparing with the performance of consecutive prediction, was evaluated on the data used for its development and internal validation. However, PIERS-ML's baseline performance has been externally evaluated (area under the receiver operating characteristic curve: 0.8), which shows very low risk of overfitting [7], and in this study we focused on the performance of consecutive prediction which was evaluated using follow-up data of the women, which was not used for training the model. While use of a completely external data set would have been preferable, the considerable deterioration over time in the most optimistic model performance is strong evidence of performance drift.

In general, we may hypothesize that static clinical prediction models can provide unreliable predictions when population characteristics, clinical practice, disease prevalence or the whole healthcare system changes—so-called "performance/calibration drift" [24–26]. For consecutive prediction using serial measurements, existing prediction models can be updated or new models trained for this purpose. Potential approaches include recalibrating intercept or both intercept and joint effects of predictors, merging previous prediction models in a meta-model, and dynamic modeling [27,28]. Dynamic modeling is the most complicated approach, with multiple methodological frameworks proposed, most commonly landmark prediction and joint modeling [29,30], but also time-dependent covariate modeling, trajectory classification, and machine learning [31]. To date, there have been attempts to construct dynamic models for predicting preeclampsia [32–34], but not for predicting the adverse outcomes of preeclampsia.

Based on our findings of performance drift (especially in terms of clinical utility) and the lack of better alternatives, we recommend using both the PIERS-ML and fullPIERS models for consecutive prediction of adverse maternal outcome in preeclampsia more cautiously as pregnancy progresses. Clinicians should interpret consecutive predictions with increasing uncertainty. To optimize maternal outcomes, future work may consider developing a dynamic approach, to account for individual trajectories so that predictions can be updated serially. The performance of these dynamic approaches can be compared with the results of this study to quantify their added value.

## Supporting information

**S1 Checklist. STROBE checklist.**
(DOCX)

**S1 Table. Components and definitions of adverse maternal outcomes and ethnicity.**
(DOCX)

**S2 Table. Predictors and outcome components of the PIERS-ML and fullPIERS models.** #: Numbers in this column refer to the number of certain outcome events occurring at any time following admission in the pooled database used in this study.
(DOCX)

**S1 Fig. Daily number of adverse outcomes (bars) and area under the receiver operating characteristic curve per day of the PIERS-ML model.** Dashed vertical lines indicate 95% confidence intervals.
(EPS)

**S2 Fig. Mean predicted probabilities of complications in the next 48 h (lines) and number of patients with measurements (bars) in uncomplicated course (blue) and adverse outcome group (pink) per day since admission using the fullPIERS model.**
(EPS)

**S3 Fig. Daily number of adverse outcomes (bars) and area under the precision-recall curve (AUC-PRC) per day of the fullPIERS model.** Dashed vertical lines indicate 95% confidence intervals. Fraction of adverse outcomes within 2 days is calculated as the number of women that would experience adverse outcomes within 2 days divided by the total number of remaining women on a certain day.
(EPS)

**S4 Fig. Daily number of adverse outcomes (bars) and area under the receiver operating characteristic curve per day of the fullPIERS model.** Dashed vertical lines indicate 95%

confidence intervals.
(EPS)

**S5 Fig. Decision curve analysis on day 0 (a), day 4 (b), day 8 (c), and day 13 (d) (fullPIERS model).** Net benefit indicates clinical usefulness of a model. At a chosen threshold probability, a higher net benefit indicates a better model. Here, we show the serial comparison between using the fullPIERS model and scenarios without a prediction model (either "treat all" or "treat none").
(EPS)

**S1 Text. Background information regarding graphic presentations.**
(DOCX)

## Acknowledgments

The authors sincerely thank all the participants who contributed to the PIERS-ML data set, and the relevant co-investigators for their collaborative spirit. Also, we thank Joost Akkermans for his preliminary work on this project.

## Author Contributions

**Conceptualization:** Wessel Ganzevoort, Laura A. Magee, Henk Groen, Peter von Dadelszen.

**Formal analysis:** Guiyou Yang, Tünde Montgomery-Csobán, Paul Murray.

**Methodology:** Tünde Montgomery-Csobán.

**Supervision:** Wessel Ganzevoort, Sanne J. Gordijn, Kimberley Kavanagh, Paul Murray, Laura A. Magee.

**Writing – original draft:** Guiyou Yang.

**Writing – review & editing:** Guiyou Yang, Tünde Montgomery-Csobán, Wessel Ganzevoort, Sanne J. Gordijn, Kimberley Kavanagh, Paul Murray, Laura A. Magee, Henk Groen, Peter von Dadelszen.

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
