## [Editor Report · Decision Letter 0]

10 Jun 2024

Dear Dr Groen, 

Thank you for submitting your manuscript entitled "Performance drift detected from consecutive prediction of adverse maternal outcomes of pre-eclampsia, using the PIERS-ML and fullPIERS models: A multi-country prospective observational study" for consideration by PLOS Medicine.

Your manuscript has now been evaluated by the PLOS Medicine editorial staff and I am writing to let you know that we would like to send your submission out for external peer review.

However, before we can send your manuscript to reviewers, we need you to complete your submission by providing the metadata that is required for full assessment. Please provide more detail on the intellectual property claim when completing the section on competing interests, and please state whether this is a financial competing interest, taking into consideration the information on our website: https://journals.plos.org/plosmedicine/s/competing-interests. 

Please login to Editorial Manager where you will find the paper in the 'Submissions Needing Revisions' folder on your homepage. Please click 'Revise Submission' from the Action Links and complete all additional questions in the submission questionnaire.

Please re-submit your manuscript within two working days, i.e. by Jun 12 2024 11:59PM.

Feel free to email me at lgaynor@plos.org if you have any queries relating to your submission.

Kind regards,

Louise Gaynor-Brook, MBBS PhD

Senior Editor

PLOS Medicine

---

## [Decision Letter · Decision Letter 1]

25 Jul 2024

Dear Dr Groen,

Many thanks for submitting your manuscript "Performance drift detected from consecutive prediction of adverse maternal outcomes of pre-eclampsia, using the PIERS-ML and fullPIERS models: A multi-country prospective observational study" (PMEDICINE-D-24-01802R1) to PLOS Medicine. The paper has been reviewed by subject experts and a statistician; their comments are included below and can also be accessed here: [LINK]

As you will see, the statistical reviewer has identified the absence of external validation of the model as a critical concern, and recommends validation of the model on an independent dataset. After discussing the paper with the editorial team and an academic editor with relevant expertise, I'm pleased to invite you to revise the paper in response to the reviewers' comments. We plan to send the revised paper to some or all of the original reviewers, and we cannot provide any guarantees at this stage regarding publication.

We ask that you submit your revision by Aug 15 2024 11:59PM. However, if this deadline is not feasible, please contact me by email, and we can discuss a suitable alternative.

Don't hesitate to contact me directly with any questions (lgaynor@plos.org). 

Best regards, 

Louise 

Louise Gaynor-Brook, MBBS PhD 

Senior Editor

PLOS Medicine

lgaynor@plos.org

Comments from the editors:

In line with the comments from the statistical reviewer (Reviewer 3), we would like to see external validation of the model using an independent dataset in a revised version of the manuscript. We would also like to see the outcomes of testing the models using follow-up data, as per Reviewer 3's comments. 

Comments from the reviewers: 

Reviewer #1: This was a prospective observational study evaluating the performance of the PIERS-ML and fullPIERS models for consecutive prediction of adverse maternal outcomes in women with preeclampsia, beyond their initial 48-hour assessment window in large cohort (8,843) of pregnancies. 

It demonstrates that both models exhibited a decline in predictive accuracy over time and were most predictive at the first assessment point within 48 hours of presentation.

The study recommends that the models are only used for initial testing. They suggest that there is a need for a more dynamic model that accommodates changing measurements and the effects of therapy. This should lead to a change in guidance. 

However, even though there is a performance drift, there is still some predictive value that can be used along with clinical assessments, especially at the extreme ends. So the models are not invalid but could be seen as a clinical support tool that could be improved by additional research using modern techniques. 

This is particularly true in resource limited settings were continuous real-time assessments may no be feasible. The PIERS models could still be useful.

Risk assessment is just that, assessing risk. the clinical care needs to be adapted by the patients's wishes and the clinical developments.

While the study provides important insights into the limitations of the PIERS-ML and fullPIERS models for consecutive predictions, these models still hold value in clinical practice. A more balanced approach that focuses on improving the models and integrating them into a broader clinical framework, rather than limiting their use, would better serve the needs of patients and clinicians.

Until further new guidance is produced of a more dynamic model, it would be premature to recommend not to use the models beyond the first assessment.

Reviewer #2: Thank you for asking me to review this interesting manuscript which addresses the value of repeated application of a predictive model to detect adverse outcome in preeclampsia. The two tools evaluated included the (original) full PIERS model and the (recent) PIERS Machine Learning model. This latter model includes more predictor variables, and has been developed- and validated- in a multiethnic population, published in Lancet Digital Health this year. It outperforms the full PIERS model, in part (i) because of it adds baseline adjustment for national GDP and MMR to the model and (ii) it has removed symptoms, with associated subjectivity. The ML PIERS model was able to stratify maternal risk, with high LR and event rate within 48 hours accompanying those classified as high or very high risk, and very low event rates among those classified as lowest risk. Knowing the risk of severe adverse outcome in the next 48 hours can be used to inform care decisions such as transfer, admin of corticosteroids etc.

The question asked in this paper is whether the PIERS model (either full PIERS or ML PIERS) can be reapplied daily during the disease course to better identify women at high risk of adverse outcome beyond the initial triage. Perhaps unsurprisingly, the performance of this model decays over time post admission. 

1. This may be, firstly, due to predictors indicating an outcome already present, eg oxygen saturation as a predictor- and oxygen saturation less than 90% as an outcome, which inflates 'predictive' power. Do you have data on what proportion of patients this applied to? 

2. The presence of some of these predictors will necessitate immediate delivery. This means the worst group- where the predictive model will perform best- is rapidly removed from the cohort. Although the cohort has been described elsewhere, some baseline demographics such as gestational age and whether patients were universally undergoing expectant management would help to interpret threshold for delivery.

3. For those remaining, intervention may also prevent the adverse outcome (eg controlling severe HT to reduce CNS sequelae). The impact of treatment/ intervention is not accounted for in the model, which will limit its applicability across time. 

4. While the decay of performance post initial triage is unsurprising, repeating the PIERS model may still have value (albeit more modest) as suggested by NICE in identifying women at risk. This would mean undertaking a prospective study comparing a formal PIERS reassessment to current 'standard of care'- repeated bedside assessment. Do you have any information on this sort of comparison? 

The methodology is very comprehensive, and the graphs are sophisticated. As one not used to interpreting these novel visual representations, I did find them- and the accompanying text- very hard to digest. Others in the general readership may also find this difficult. This may benefit from examples or detailed explanations in the supps to improve readability and accessibility of the paper.

Specific questions: 

1) Given the miniPIERS model was initially developed for LMIC, why did you not compare the performance of this model as well?

2) The cohorts should be better described in terms of years of recruitment, gestational age etc. Were these patients all preterm undergoing expectant management (line 101)? Information about GA helps to interpret the rapid fall in number of patients providing measurements across time. 

3) is there a reason for not using the iHOPE core outcome set which includes perinatal outcomes? I presume this is because you didn't have the variables collected given some of these cohorts dated back to 2003. Please clarify. 

4) Can you please explain why you think the fullPIERS model performed so poorly at baseline (Fig S2) compared to ML PIERS (Fig 1)

5) This paper assumes quite a lot of knowledge, particularly from the Lancet Digital Health paper. I would suggest at least providing an example of a Precision Recall plot so the readership can understand how the summary AUCs have been generated that are subsequently presented in Fig 2 and Fig S3. Also it would help if the y axis was identical in these 2 graphs so the different performance of full PIERS and ML PIERS was immediately visible.

6) The decision curve analyses are quite complex to follow. Further, it's said that the model is better than treat all for prob of 3.1% on Day 0, but this isn't clearly apparent in the figure (referenced Fig 3b, but should be fig 3a, I think). Treat all is better for this group on Day 4 (Fig 3b). The statement that these 'align with previous findings' is not referenced (line 317). 

Reviewer #3: "Performance drift detected from consecutive prediction of adverse maternal outcomes of pre-eclampsia, using the PIERS-ML and fullPIERS models: A multi-country prospective observational study" examines the existing Pre-eclampsia Integrated Estimate of RiSk (PIERS) models for ongoing assessment beyond 48 hours, as the PIERS models had originally only been developed and validated as "static" models as part of an initial assessment within 48 hours. The multi-country prospective observational study assessed both PIERS-ML and fullPIERS models for each day within a two-week post-admission window, and found that while PIERS-ML had acceptable performance for specific risk groups, this was not the case for other risk groups beyond 48 hours. As such, it was recommended to use the PIERS models only for the first day, as originally intended and validated.

1. In the Data Source subsection, it is stated that "The baseline data of this pooled database... was used for development of the PIERS-ML model". From [7], PIERS-ML was developed for outcomes within 2 days (and 7 days), using clinical data at admission (baseline data)

From what could be understood, "consecutive/ongoing prediction" would then be applying the PIERS-ML model developed only on the baseline data, to follow-up measurements (i.e. for say the 7th day post-admission, treat the clinical values from the 7th day as the baseline, to predict up to the 9th day for the PIERS-ML 2-day model). If this is correct, the perhaps obvious approach would seem to be to develop (and validate) separate PIERS-ML models for consecutive/ongoing prediction with the follow-up data (i.e. for each day as baseline); it is then curious as to why this does not appear to have been considered/discussed.

2. While performance drift was detected, this does not seem very surprising as-is, since one might expect outcomes of pre-eclampsia to vary (significantly) over time from admission. As such, while the analysis on existing static models may be instructive, the practical contribution of the study is less clear.

3. In the Strengths and Limitations section, a limitation was stated as PIERS-ML and fullPIERS being partly evaluated on baseline data used for development. This appears to be a fairly critical concern pertaining to accurate evaluation, and a more convincing evaluation would be to perform the analysis only on test data not involved in the development of PIERS-ML/fullPIERS, as well as additional external evaluation data.

---

* Please upload any figures associated with your paper as individual TIF or EPS files with 300dpi resolution at resubmission; please read our figure guidelines for more information on our requirements: http://journals.plos.org/plosmedicine/s/figures. While revising your submission, please upload your figure files to the PACE digital diagnostic tool, https://pacev2.apexcovantage.com/. PACE helps ensure that figures meet PLOS requirements. To use PACE, you must first register as a user. Then, login and navigate to the UPLOAD tab, where you will find detailed instructions on how to use the tool. If you encounter any issues or have any questions when using PACE, please email us at PLOSMedicine@plos.org.

* Please note that the Data Availability Statement as currently written does not meet with PLOS requirements. 

FIGURES AND TABLES

SUPPLEMENTARY MATERIAL

REFERENCES

MODELLING STUDIES

The following list is derived from Geoffrey P Garnett, Simon Cousens, Timothy B Hallett, Richard Steketee, Neff Walker. Mathematical models in the evaluation of health programmes. (2011) Lancet DOI:10.1016/S0140-6736(10)61505-X: 

* If pertinent, please provide a diagram that shows the model structure, including how the natural history of the disease is represented, the process and determinants of disease acquisition, and how the putative intervention could affect the system.

* Please provide a complete list of model parameters, including clear and precise descriptions of the meaning of each parameter, together with the values or ranges for each, with justification or the primary source cited and important caveats about the use of these values noted.

* Please provide a clear statement about how the model was fitted to the data, including goodness-of-fit measure, the numerical algorithm used, which parameter varied, constraints imposed on parameter values, and starting conditions.

* For uncertainty analyses, please state the sources of uncertainties quantified and not quantified [can include parameter, data, and model structure].

* Please provide sensitivity analyses to identify which parameter values are most important in the model. Uncertainty estimates seek to derive a range of credible results on the basis of an exploration of the range of reasonable parameter values. The choice of method should be presented and justified.

* Please discuss the scientific rationale for the choice of model structure and identify points where this choice could influence conclusions drawn. Please also describe the strength of the scientific basis underlying the key model assumptions.

* For studies that develop a prediction model or evaluate its performance, please ensure that the study is reported according to the TRIPOD statement (https://www.equator-network.org/reporting-guidelines/tripod-statement) and include the completed checklist as Supporting Information. Please add the following statement, or similar, to the Methods: "This study is reported as per the Transparent Reporting of a Multivariable Prediction Model for Individual Prognosis Or Diagnosis (TRIPOD) statement (S1 Checklist)." For studies using machine learning, please use the TRIPOD-AI checklist. When completing the checklist, please use section and paragraph numbers, rather than page numbers.

---

## [Decision Letter · Decision Letter 2]

1 Nov 2024

Dear Dr. Groen,

Thank you very much for re-submitting your manuscript "Performance drift detected from consecutive prediction of adverse maternal outcomes of pre-eclampsia, using the PIERS-ML and fullPIERS models: A multi-country prospective observational study" (PMEDICINE-D-24-01802R2) for review by PLOS Medicine.

I have discussed the paper with my colleagues and the academic editor and it was also seen again by three reviewers. I am pleased to say that provided the remaining editorial and production issues are dealt with we are planning to accept the paper for publication in the journal.

[LINK]

If you have any questions in the meantime, please contact me (lgaynor@plos.org) or the journal staff on plosmedicine@plos.org.  

We look forward to receiving the revised manuscript by Nov 08 2024 11:59PM.   

Sincerely,

Louise Gaynor-Brook, MBBS PhD

Senior Editor 

PLOS Medicine

plosmedicine.org

Requests from Editors:

General comments:

Please note that line numbers indicated below refer to the .pdf version of the manuscript.

Throughout the paper, please adapt reference call-outs to the following style: "... (e.g., <34 weeks gestation) [1,2]." (noting the absence of spaces within the square brackets, and placement of the call-out before punctuation).

Please replace all instances of "subject" with participant, patient, individual, or person.

To help us extend the reach of your research, please provide any Twitter handle(s) that would be appropriate to tag, including your own, your coauthors’, your institution, funder, or lab.

Data availability:

PLOS Medicine requires that the de-identified data underlying the specific results in a published article be made available, without restrictions on access, in a public repository or as Supporting Information at the time of article publication, provided it is legal and ethical to do so. 

The Data Availability Statement (DAS) requires revision. If the data are not freely available, please describe briefly the ethical, legal, or contractual restriction that prevents you from sharing it. Please also include an appropriate, independent contact (web or email address) for inquiries - preferably a general email address rather than a specific individual. Please note that a study author cannot be the contact person for the data.

Title: Please revise your title according to PLOS Medicine's style. Your title must be nondeclarative. We suggest “Consecutive prediction of adverse maternal outcomes of pre-eclampsia using the PIERS-ML and fullPIERS models: A multi-country prospective observational study” or similar

Abstract Methods and Findings:

Please ensure that all numbers presented in the abstract are present and identical to numbers presented in the main manuscript text.

Please provide brief demographic details of the study population (e.g. age, gestational age, ethnicity and/or other important variables)

Please include the years during which the data were collected.

Please quantify the main results.

Line 45 - please clarify what is meant by ‘clinical impact’ (noting that causative language such as ‘impact’ should be avoided where possible)

Line 48 - Please be more specific regarding what is considered a low AUC-PRC value, and briefly expand upon what this means in terms of model performance. 

In the last sentence of the Abstract Methods and Findings section, please describe 2-3 of the main limitations of the study's methodology.

Abstract Conclusions:

Please begin your Abstract Conclusions with "In this study, we observed ..." or similar, to summarize the main findings from your study, without overstating your conclusions. Please emphasize what is new and address the implications of your study, being careful to avoid assertions of primacy. 

Please remove keywords.

Author Summary:

In the final bullet point of ‘What Do These Findings Mean?’, please describe the main limitations of the study in non-technical language.

Methods:

Did your study have a prospective protocol or analysis plan? Please state this (either way) early in the Methods section. If a prospective analysis plan (from your funding proposal, IRB or other ethics committee submission, study protocol, or other planning document written before analyzing the data) was used in designing the study, please include the relevant prospectively written document with your revised manuscript as a Supporting Information file to be published alongside your study, and cite it in the Methods section. A legend for this file should be included at the end of your manuscript. If no such document exists, please make sure that the Methods section transparently describes when analyses were planned, and if/when reported analyses differed from those that were planned. Changes in the analysis-- including those made in response to peer review comments-- should be identified as such in the Methods section of the paper, with rationale. If a reported analysis was performed based on an interesting but unanticipated pattern in the data, please be clear that the analysis was data-driven.

Please incorporate your ethics statement (lines 425-6) into the Methods section. Please specify whether informed consent for participation in the study was written or oral.

Please ensure that the study is reported according to the STROBE guideline, and include the completed STROBE checklist as Supporting Information. Please add the following statement, or similar, to the Methods: "This study is reported as per the Strengthening the Reporting of Observational Studies in Epidemiology (STROBE) guideline (S1 Checklist)." The STROBE guideline can be found here: http://www.equator-network.org/reporting-guidelines/strobe/ When completing the checklist, please use section and paragraph numbers, rather than page numbers which will likely no longer correspond to the appropriate sections after copy-editing.

Lines 167, 168 - please consider revising to ‘potential clinical impact’ given that this refers to a model.

Results: 

Please provide a table (as Table 1) showing the baseline characteristics of the study population, including the number of participants from each country/region represented.

Line 201 - please clarify that this refers to the mean predicted probability of adverse outcomes in the adverse outcome group.

Line 205 - please clarify that this refers to the mean predicted probability of adverse outcomes in the uncomplicated-course group.

Line 215 - please quantify the AUC-PRC values on day 0 and day 1 as, by eye, the day 1 value does not appear to be similar to day 0 and is below 0.6.

Lines 247-257 - please be clear that your results are predicted e.g. “treatment guided by the model was predicted to be as good as treating all patients”.

Line 272 - please specify the timeframe i.e. whether this is also at risk at least once in the two weeks from after first assessment.

Discussion:

Please present and organize the Discussion as follows: a short, clear summary of the article's findings; what the study adds to existing research and where and why the results may differ from previous research; strengths and limitations of the study; implications and next steps for research, clinical practice, and/or public policy; one-paragraph conclusion.

Please remove all subheadings within your Discussion e.g. Key results.

Line 359 - would suggest revising to “...altered its accuracy and contributed to its inferiority…”

Figures:

Please consider avoiding the use of red and green in order to make your figure more accessible to those with colour blindness.

Fig 3 legend - please add ‘respectively’ to meaning of threshold possibilities. 

Fig 4 - please provide a little more detail in the figure legend for how to interpret this figure, in addition to S1 Text.

Tables:

Table 1 - please indicate what is represented by the numbers in brackets. 

When a p value is given, please specify the statistical test used to determine it in the table legend.

Please define all abbreviations used in the table legend of each table.

References:

Please ensure that journal name abbreviations match those found in the National Center for Biotechnology Information (NCBI) databases (http://www.ncbi.nlm.nih.gov/nlmcatalog/journals), and are appropriately formatted and capitalised.

Please also see https://journals.plos.org/plosmedicine/s/submission-guidelines#loc-references for further details on reference formatting. 

Where website addresses are cited, please specify the date of access. 

Comments from Reviewers:

Reviewer #1: I think this is a far more balanced paper highlighting the fall off of predictive value but acknowledging the continued (cautious) use in repeated evaluations. The authors have argued the reviewers fairly and responded to the comments made.

Reviewer #2: Thank you very much for your comprehensive response to my questions. You have addressed nearly all my queries. 

My one residual question relates (again) to the decision curve analyses. You have corrected my initial comment re misnaming 3a and 3b, but in this revised manuscript line 249-251 says that, 'when the threshold probability was 3.1%, treatment guided by the model only yielded higher benefit (than treating all) on day 4 (Fig 3b). Yet Fig 3b still seems to show benefit for 'treat all' compared to the model at the 3.1% threshold. Can you please clarify? 

Reviewer #3: We thank the authors for addressing our previous concerns.

[LINK]

---

## [Editor Report · Decision Letter 3]

5 Dec 2024

Dear Dr Groen, 

On behalf of my colleagues and the Academic Editor, Prof. Gordon Smith, I am pleased to inform you that we have agreed to publish your manuscript "Consecutive prediction of adverse maternal outcomes of pre-eclampsia, using the PIERS-ML and fullPIERS models: A multi-country prospective observational study" (PMEDICINE-D-24-01802R3) in PLOS Medicine.

PRESS

Sincerely, 

Louise Gaynor-Brook, MBBS PhD 

Senior Editor 

PLOS Medicine